# Purification and Characterization of Immunoglobulin Y (IgY) Targeting Surface Antigen 1 (SAG1) of *Toxoplasma gondii*

**DOI:** 10.3390/antib14040081

**Published:** 2025-09-26

**Authors:** Enrique Adrián Herrera-Aguirre, Diana León-Núñez, Jaime Marcial-Quino, Saúl Gómez-Manzo, César Augusto Reyes-López, Yolanda Medina-Flores, Olga Mata-Ruíz, Lizbeth Xicotencatl-García, Hector Luna-Pastén, Luz Belinda Ortiz-Alegría, Nury Pérez-Hernández, Magdalena Escorcia, Dolores Correa, Fernando Gómez-Chávez

**Affiliations:** 1Departamento de Medicina y Zootecnia de Aves, Facultad de Medicina Veterinaria y Zootecnia, Universidad Nacional Autónoma de México, Mexico City 04510, Mexico; mvzeaha@gmail.com (E.A.H.-A.); magdaesc@unam.mx (M.E.); 2Laboratorio de Enfermedades Osteoarticulares e Inmunológicas, Sección de Estudios de Posgrado e Investigación, Escuela Nacional de Medicina y Homeopatía (IPN), Mexico City 07320, Mexico; dianaleonn1300@egresado.ipn.mx (D.L.-N.); nperezh@ipn.mx (N.P.-H.); 3Laboratorio de Inmunología Experimental, Instituto Nacional de Pediatría, Mexico City 04530, Mexico; hec_vene@hotmail.com (H.L.-P.); bel_alegria@yahoo.com.mx (L.B.O.-A.); 4Laboratorio de Biología y Quimica, Plantel Casa Libertad, Universidad Autónoma de la Ciudad de México, Mexico City 09620, Mexico; jmarcialq@ciencias.unam.mx; 5Laboratorio de Bioquímica Genética, Instituto Nacional de Pediatría, Mexico City 04530, Mexico; saulmanzo@ciencias.unam.mx; 6Laboratorio de Bioquímica Estructural, Sección de Estudios de Posgrado e Investigación, Escuela Nacional de Medicina y Homeopatía (IPN), Mexico City 07320, Mexico; careyes@ipn.mx; 7Laboratorio de Anticuerpos Monoclonales, Departamento de Biología Molecular y Validación de Técnicas, Instituto de Diagnóstico y Referencia Epidemiológicos Dr. Manuel Martínez Báez (InDRE), Mexico City 01480, Mexico; yolanda.medina@salud.gob.mx (Y.M.-F.); olga.mata@salud.gob.mx (O.M.-R.); 8Bioterio, Instituto Nacional de Pediatría, Mexico City 04530, Mexico; lizxico@yahoo.com.mx; 9Centro de Investigación en Ciencias de la Salud, Facultad de Ciencias de la Salud, Universidad Anáhuac México, Av. Universidad Anáhuac 46, Lomas Anáhuac, Huixquilucan 52786, Estado de México, Mexico; dolores.correa@anahuac.mx

**Keywords:** *Toxoplasma gondii*, SAG1, IgY, antibodies

## Abstract

*Toxoplasma gondii* (*T. gondii*) is an obligate intracellular protozoan parasite responsible for toxoplasmosis, a disease with significant health implications for humans and animals. The surface antigen 1 (SAG1) of *T. gondii* is a major immunodominant protein that facilitates host cell invasion, making it an ideal target for diagnostic and therapeutic interventions. Immunoglobulin Y (IgY), the primary antibody in avian species, offers unique advantages over mammalian IgG, including easier animal care, lower costs, high-yield production, and potential passive immunization. Objectives: This study aimed to induce, purify, and characterize IgY antibodies targeting *T. gondii* SAG1 from hen egg yolks. Methods: The coding region of the mature portion of *T. gondii* SAG1 was amplified by PCR, cloned into the pET32a(+) vector for heterologous expression in *E*. *coli*. The recombinant SAG1 (rSAG1) was purified by affinity chromatography and used to immunize hens. IgY was extracted from egg yolks using PEG. SDS-PAGE and spectrophotometry were used to evaluate purity and concentration. By ELISA, Western blot, and flow cytometry, the specificity of IgY was assessed against recombinant and endogenous, native, and denatured SAG1. Results: Purified IgY demonstrated strong recognition of both recombinant and native SAG1 in ELISA and Western blot, and against *T. gondii* tachyzoites by flow cytometry. Conclusions: SAG1-specific IgY was produced in a pure form; it could be helpful in research, diagnosis, and treatment at low costs on a larger production scale, with minimal animal harm.

## 1. Introduction

Toxoplasmosis is an infection caused by the parasite *Toxoplasma gondii* (*T. gondii*). This neglected parasitosis can affect many intermediate hosts, including humans [1,2,3,4]. Approximately one-third of the world’s population is estimated to be chronically infected with the parasite. In the specific case of Mexico, the adjusted prevalence ranged from 27.9% to 43.9% from 1951 to 2022 [5,6,7]. In immunocompetent individuals, toxoplasmosis is asymptomatic in more than 80% of cases. In contrast, in people with compromised immune systems, the infection can trigger fever, dizziness, pneumonia, seizures, and even severe complications such as encephalitis, respiratory failure, kidney failure, septic shock, and, in extreme cases, death [1,8]. *T. gondii* can be transmitted horizontally through contaminated food and water or vertically from an infected pregnant woman to her fetus, leading to congenital toxoplasmosis (CT), which is a significant health issue, with an estimated rate of two cases per 1000 births in Mexico City [9]. The infection can cause severe fetal complications, including hydrocephalus, intracranial calcifications, deafness, intellectual disability, seizures, retinochoroiditis, blindness, and even abortion or fetal/neonatal death [2,7,8,9]. These challenges hinder effective treatment and management, especially in preventing fetal transmission. Pregnant women, newborns, and infants can be treated with a combination of antibiotics such as pyrimethamine and sulfadiazine or spiramycin, along with folinic acid; however, *T. gondii* is not eliminated [8,10,11]. Therefore, it is considered that there is no effective and definitive treatment for toxoplasmosis. This ensures fertile ground for developing new treatments against this parasite [12]. Diagnosing CT also remains challenging. For example, the timing of maternal infection and the immune responses of both the mother and the fetus can influence the development of the condition [13,14,15]. Many pregnant women are asymptomatic, leading to misdiagnosis. In some cases, it is further complicated by the persistence of the mother’s IgM antibodies, which make it difficult to distinguish between acute and chronic infections and assess the risk of vertical transmission [16]. Additionally, actual CT diagnostic tests’ varying sensitivities and specificities complicate precision, emphasizing the need for comprehensive testing, such as capture ELISA and recombinant proteins, which can increase performance [17].

SAG1, also known as p30, plays a crucial role in *T. gondii* for cell recognition, binding, and invasion [18]. This protein has allelic forms, although its exposed domains are conserved among *T. gondii* strains [19]. It has been used for research and diagnosis, serving as a target for T and B lymphocytes, making it a candidate for vaccination or antibody treatment [20,21,22,23,24,25,26,27,28]. Polyclonal and monoclonal antibodies have been produced and utilized as essential components of IgM and IgG anti-parasite detection in commercial kits based on indirect and capture ELISA [29,30]. Additionally, antibodies have been utilized for CT treatment in mice [28]. However, their production either involves animal suffering or costly procedures.

Immunoglobulin Y technology enables large-scale production of specific antibodies while adhering to international bioethical guidelines, which minimize animal suffering and maintain refined procedures. These antibodies can be either polyclonal or monoclonal, targeting conserved mammalian proteins with minimal cross-reactivity in diagnostic applications, as the rheumatoid factor does not recognize them [31,32,33,34,35]. Moreover, their inability to activate the complement system enhances their suitability for therapeutic and prophylactic purposes. These attributes, along with their adaptability, make IgY antibodies valuable diagnostic and biotherapeutic tools [36].

This study focused on the induction, purification, and characterization of SAG1-specific IgY antibodies against *T. gondii,* which could be a potential tool for further development in diagnostic platforms and exploratory therapeutic research targeting this parasite.

## 2. Materials and Methods

### 2.1. Bioethics

The procedures of this study were conducted in accordance with the Mexican guidelines for animal experimentation (Mexican Official Standard, NOM-062-ZOO-1999 [37]), which adhere to the principles of the 3Rs. The euthanasia procedure for mice was cervical dislocation, which is allowed by international standards for animals weighing <200 g [38]. This work was approved by the research committee of the Instituto Nacional de Pediatría (2020/025, 1 June 2020) and the ethics committee (2022/0678, 7 January 2022) of the Facultad de Medicina Veterinaria Y Zootecnia, UNAM.

### 2.2. T. gondii Tachyzoites for Cytometry and Crude Extract Preparation

The *T. gondii* RH strain (type I) was used in this study. The parasite was maintained through passages in Balb/c mice according to a previously reported protocol, which included euthanasia of mice that showed disease-related signs [39]. Briefly, the mice received 10 tachyzoites via i.p. injection, and after three days, the parasites were harvested from the peritoneum. The animals were then euthanized by cervical dislocation. A crude extract (TgCE) was prepared as previously reported [39]. Briefly, the tachyzoites were isolated from mouse peritoneum, lysed with wide-spectrum proteinase inhibitors, and ultracentrifuged; the supernatant was frozen at −80 °C and aliquoted until use.

### 2.3. Recombinant SAG1 (rSAG1) and E. coli Crude Extract

As SAG1 doesn’t contain introns [40], DNA from *T. gondii* RH tachyzoites was isolated following the method described by Homan et al. (2000) [41]. Genomic DNA was extracted using the Qiagen Gentra^®^ Puregene^®^ Tissue Kit (Hilden, Germany), following the manufacturer’s instructions. The isolated DNA was quantified using a NanoDrop™ 1000 spectrophotometer (Thermo Scientific, Waltham, MA, USA) and stored at −20 °C until further use.

The coding sequence for the mature form of *T. gondii* SAG1 (residues 78–306) was amplified by PCR [20]. One hundred ng of DNA and the Phusion™ High-Fidelity DNA Polymerase enzyme were used to amplify the fragment. The primers used were SAG1 Fw: TAATCCATGGAGAACCACTTCACTCT and SAG1 Rv: GGGCTGCAGGGCTCGAGTAT. Amplification was performed in a H100 thermal cycler (Bio-Rad, Hercules, CA, USA) under the following conditions: 30 s at 98 °C; 30 cycles (10 s at 98 °C, 30 s at 60 °C, and 30 s at 72 °C); and finally, 2 min at 72 °C. The NcoI and XhoI restriction sites (underlined) were added to the primers for cloning the fragment into the pET-32a(+) expression vector (Novagen, Madison, WI, USA; now MilliporeSigma) [42]. The recombinant plasmid pET-32a(+)-SAG1 was transformed into *Escherichia coli* BL21(DE3) for rSAG1 production or *E. coli* without a plasmid for crude extract production. Both transformed and non-transformed *E. coli* were cultured overnight in a Luria Broth medium (50 mL) supplemented with ampicillin (100 μg/mL). The culture was transferred to 2 L of Luria Broth medium containing ampicillin (100 μg/mL) and incubated at 37 °C and 180 rpm until an optical density at 600 nm (OD600) of 0.5 was reached. At this stage, protein expression was induced with 0.5 mM isopro-pyl-β-D-thiogalactopyranoside (IPTG). After 18 h of induction, the cells were harvested by centrifugation (4000× *g* for 20 min at 4 °C) and the pellet obtained was stored at −20 °C until use. The cell pellets were resuspended in a lysis buffer with the following composition: 100 mM Tris (pH 8.0), 50 mM NaCl, 5 mM mercaptoethanol, and 1 mM phenylmethylsulfonyl fluoride (PMSF). The cells were then sonicated for 8 cycles (45 s with 1 min rest periods) and subsequently centrifuged for 20 min at 15,000× *g* and 4 °C to obtain the enzymatic extract used for protein purification [43,44]. Briefly, the supernatant was subjected to nickel affinity chromatography (Ni-NTA) to purify the 6 × His rSAG1 protein and analyzed by 12% SDS-PAGE. The purified rSAG1 was collected for hen immunization.

### 2.4. Hen Immunization and Egg Collection

Ten 50-week-old ALPES hens were housed for two weeks before hyperimmunization to allow blood catecholamine levels to return to baseline and estrogen levels to stabilize. Once acclimated, the hens were immunized with the recombinant *T. gondii* SAG1 (rSAG1) protein.

Immunization consisted of three injections into the breast muscle, each containing 150 μg of the rSAG1 antigen in 1% chitosan as an adjuvant, in a final volume of 0.2 mL. The second dose was administered two weeks after the first, followed by a third dose one week later.

From the first week of immunization, all eggs laid by the hens were collected until the study concluded, 23 days after the final inoculation. A total of 152 eggs were collected and stored at 4 °C until processing.

### 2.5. Purification of IgY from Egg Yolks Using Polyethylene Glycol (PEG) 6000

IgY purification from the yolks of eggs from hyperimmunized hens was performed following the method described by Pauly et al. (2011) [45]. The egg yolk was transferred into a 50 mL conical tube and mixed with a 3.5% (*w*/*v*) solution of Polyethylene Glycol 6000 (PEG 6000) (MilliporeSigma, Burlington, MA, USA) in phosphate-buffered saline (PBS, pH 7.2). Two additional precipitation steps were performed using PEG 6000 at final concentrations of 8.5% and 12%. The purified IgY was analyzed by SDS-PAGE under native conditions using the 4–15% Mini-PROTEAN^®^ TGX™ Precast Protein Gels (Bio-Rad, Hercules, CA, USA). The IgY antibodies were dialyzed overnight against PBS and stored at −20 °C until use.

### 2.6. ELISA for IgY Anti-SAG1

For the detection of specific anti-SAG1 IgY antibodies, an indirect ELISA was standardized on an ad hoc basis. The plates were coated with 2 μg/mL of *T. gondii* RH strain crude extract (TgCE), as previously standardized [46], or with rSAG1 in concentrations ranging from 7.8 ng/mL to 8.0 μg/mL. Antigens were diluted in 0.01 M carbonate buffer (pH 9.6) and incubated overnight at 4 °C. After coating, the plates were washed three times with PBS-0.1% Tween 20 (PBS-T), followed by the addition of 200 µL of PBS-T containing 1.0% BSA (T-BSA) as a blocking agent, and incubated for 30 min. Then, the plates were washed again, as described above, and 100 μL of purified IgY antibodies (serially diluted in PBS-T from 20 μg/mL to 0.3125 μg/mL) were added to each well. The plates were incubated for 1.5 h at 37 °C. After another round of washes, 100 µL/well of rabbit anti-chicken IgY (whole molecule) conjugated with horseradish peroxidase (Dil 1:2500, Thermo Scientific, Wilmington, DE, USA, Cat. No A9046) in PBS-T was added. The plates were incubated for 1.5 h. Finally, 100 µL/well of the chromogen solution, containing 4.0 mg of O-phenylenediamine and 0.2% H_2_O_2_ dissolved in 5.0 mL of 0.1 M citric acid/0.1 M sodium citrate, was added and allowed to develop for up to 30 min in the dark. The reaction was stopped by adding 50 µL/well of 0.1 M sulfuric acid. The absorbance values were obtained in a plate reader (9300-010 Modulus, Turner BioSystems, Sunnyvale, CA, USA) at 490 nm wavelength and captured with the Modulus Microplate Reader program. Then, we followed Beatty et al.’s method for the dissociation constant (Kd) measurement [47].

### 2.7. IgY Anti-SAG1 Detection by Western Blot

The purified IgY or TgCE was resolved using SDS-PAGE under native and denatured conditions, respectively, with 4–15% Mini-PROTEAN TGX Precast Protein Gels and then transferred to nitrocellulose membranes (Bio-Rad, Hercules, CA, USA). At room temperature, nonspecific binding was blocked by incubating the membranes in 5% nonfat dry milk in PBS-T for 1 h. After three 5 min washes with PBS-T for total IgY detection, the membrane was incubated with the specific rabbit anti-IgY HRP (diluted 1:5000) for 1.5 h at 37 °C. For specific anti-SAG1 IgY, the membrane was incubated with the purified anti-SAG1 IgY (10 µg/mL) for 1.5 h at 37 °C. The membrane was rewashed and incubated with the specific rabbit anti-IgY HRP (diluted 1:5000) for 1.5 h at 37 °C. Following additional washes with PBS-T, the specific bands were detected using chemiluminescence with the ECL Plus Western Blotting Detection kit (Amersham Pharmacia Biotech, Uppsala, Sweden), and the results were captured using the Gel Doc XR+ Gel Documentation System (Bio-Rad, Hercules, CA, USA).

### 2.8. IgY Anti-SAG1 Alexa Fluor 488 Labeling and SAG1 Detection in Tachyzoites by Flow Cytometry

Purified IgY anti-SAG1 at a concentration of 2 mg/mL was fluorescently labeled using the Labeling kit for Alexa Fluor 488 (Invitrogen, Carlsbad, CA, USA) according to the manufacturer’s instructions. The Alexa Fluor 488 IgY anti-SAG1 diluted 1:10, 1:100, and 1:1000 was used to stain 30 × 10^3^ paraformaldehyde-fixed RH *T. gondii* tachyzoites suspended in PBS by incubation for 30 min. The samples were analyzed using a Guava easyCyte 6-2L flow cytometer (Merck Millipore, Burlington, MA, USA).

## 3. Results

### 3.1. Recombinant SAG1 (rSAG1) Production in E. coli

To induce IgY production in hens, we produced recombinant SAG1 mature protein from RH *T. gondii* in *E. coli* using the pET32a(+) plasmid, which is used to express recombinant proteins fused to thioredoxin to facilitate their folding [47]. The predicted recombinant fusion protein (rSAG1), with an expected molecular weight of 44 kDa, was highly expressed in *E. coli* following IPTG induction and could be efficiently purified by affinity chromatography, as revealed by SDS-PAGE (Figure 1).

### 3.2. Anti-SAG1 IgY Production, Purification from Egg Yolks, and Characterization

IgY antibodies against rSAG1 were induced in hens and purified, as demonstrated by SDS-PAGE (Figure 2A) and Western blot (Figure 2B). The average concentration of IgY per egg before and after the first immunization was 5.57 mg/mL. A noticeable increase was seen after the second and third immunizations, reaching an average concentration of 10.15 mg/mL. This gradual rise is consistent with the expected patterns and concentrations of IgY production in laying hens, as reported elsewhere [48,49]. This confirms the high purity of IgY, with the expected 150 kDa molecular weight of IgY antibodies.

### 3.3. Anti-SAG1 IgY Can Detect Recombinant SAG1 (rSAG1) from RH T. gondii

Once we confirmed the efficiency of IgY purification, we assessed its specific detection of rSAG1 during immunization of hens. We found that specific anti-rSAG1 IgY increased following the first immunization (Figure 3A). We also determined its detection limit using an indirect ELISA, discovering that the detection range of IgY concentration at the end of the hens’ immunization was between 0.3125 and 20 μg/mL, which could detect rSAG1 at concentrations as low as 0.25 μg/mL (Figure 3B). Furthermore, the dissociation constant (Kd) was calculated following the method described by Beatty et al., yielding a value of 2.22 × 10^−9^ M, which indicates a high-affinity interaction between IgY anti-SAG1 and SAG1.

### 3.4. Anti-SAG1 IgY Can Detect Native SAG1 Present in the RH T. gondii Crude Extract

To evaluate whether anti-SAG1 IgY induced by rSAG1 could detect both native and denatured endogenous *T. gondii* SAG1, we used the crude extract from RH *T. gondii* tachyzoites (TgCE) as antigen using indirect ELISA (Figure 4A). We found that IgY anti-SAG1 can detect the native SAG1 present in TgCE (2 μg/mL) in an IgY concentration-dependent manner. Additionally, we found that IgY anti-SAG1 can detect denatured SAG1 in the TgCE (Figure 3B).

### 3.5. Anti-SAG1 IgY Can Detect Native SAG1 on the Surface of RH Tachyzoites

As shown in Figure 5, Alexa Fluor 488-conjugated anti-SAG1 IgY detected SAG1 on paraformaldehyde-fixed RH tachyzoites by flow cytometry, with MFI values increasing with higher antibody concentrations: 97 at 1:100 and 390 at 1:10 dilution. These results confirm the specific recognition of SAG1 by anti-SAG1 IgY in tachyzoites.

## 4. Discussion

Toxoplasmosis remains a global health concern, particularly for immunocompromised individuals and pregnant women [50]. This highlights the need for enhanced diagnostic tools and improved treatment options. Serological assays are the most common methods for diagnosing *T. gondii* infection, emphasizing the need for well-characterized immunological reagents [17].

Several antibodies against parasite antigens have been produced, including monoclonals with SAG1 specificity [51]. However, these reagents required the use of mammals, primarily mice, leading to significant suffering during ascites induction. Alternatively, clones are expanded in vitro, which incurs high costs due to the use of culture media and the need to purify the molecules using sophisticated affinity-based chromatographic columns [51,52]. In this study, we produced, characterized, and purified IgY antibodies that target SAG1 of RH *T. gondii* as an immunoreactive alternative source to antibodies from mammals. The results demonstrated that specific IgY anti-SAG1 can be produced in hens, purified from egg yolks, and used to detect denatured, native, recombinant, purified SAG1 from *E. coli* and TgCE, as well as directly on fixed tachyzoites.

Using hens for antibody production aligns with ethical guidelines emphasizing animal welfare. This method reduces the distress typically associated with traditional practices. Chicken IgY antibodies are obtained from egg yolks, thereby avoiding the painful blood collection process from animals, which is inherently distressing to them. Additionally, hens can produce a significantly higher number of antibodies than mammals, thereby reducing the number of animals required for long-term antibody production. This approach adheres to the 3Rs principles of animal research (Replacement, Reduction, and Refinement), providing a more humane and efficient alternative [36,53,54].

Furthermore, IgY antibodies offer several benefits, including high yield, ease of purification, and reduced immunogenicity in mammals. In addition, they do not bind to mammalian Fc receptors, rheumatoid factor, or complement proteins (C1q and C3). These characteristics help prevent false-positive results in diagnostic research and underscore the growing demand for IgY in therapeutic areas where anaphylactic reactions are unavoidable [36,53,54].

Since 1987, Hasasl et al. have attempted to replace rabbit IgG with yolk IgY antibodies in their homemade ELISA for parasite detection, producing IgY anti-*T. gondii* crude extract of the BK strain. Nevertheless, they found differing immunoprecipitation patterns among mammalian and hen antibodies, arguably due to species differences in recognizing immunodominant antigenic determinants in BK TgCE. Therefore, they consider it unable to shift the specificity and reliability of its serological test, highlighting the importance of using purified or recombinant antigens [55]. IgY anti-TgCE antibodies have also been reported as suitable for detecting various forms of *T. gondii* in cultured cell monolayers and paraffin-embedded mouse tissues. These antibodies also displayed a differentially recognized profile against antibodies produced in mice. Still, they did not exhibit cross-reactivity with other apicomplexans, remarking that polyclonal IgY antibodies are valuable tools for studies involving *T. gondii* [56].

On the other hand, SAG1 is an immunodominant protein essential for the parasite’s capacity to infect host cells [18]. It is highly conserved across various strains, making it eligible for diagnosis, treatment, and understanding the parasite’s biology. Its conservation can be observed by examining the *SAG1* gene sequence, which exhibits minimal variation among isolates. Additionally, it plays an essential role in the interaction with host cells and thus in the cell cycle of the parasite [57]. Due to the importance of SAG1 in *T. gondii*’s biology, different efforts have been conducted to produce IgY against this immunodominant antigen [58,59,60]. The first attempt to produce IgY antibodies against SAG1 was made by sonicating *T. gondii* tachyzoites, but not by purifying or producing them recombinantly. The profile analysis of the membrane proteins obtained from sonicated tachyzoites revealed predominantly proteins with molecular weights of approximately 35, 60, 66, 81, and 86 kDa. IgY from egg yolk mainly recognized *T. gondii* protein antigens with molecular weights of around 36 kDa and 79 kDa by Western blot. The authors suggested that the 36 kDa molecule would be SAG1, as this antigen is the major surface protein on RH *T. gondii* tachyzoites; however, further studies to test the specificity of IgY anti-sonicated soluble proteins from RH *T. gondii* tachyzoites were not done [58]. The group of Cakir-Koc et al. has also produced IgY anti-SAG1 for its use in ELISA [59] and IF [60]. Still, in contrast to these reports, where they employed a commercial recombinant protein consisting of the 45–198 amino acid region of SAG1, we employed an in-house-made SAG1 mature form expressed in the pET32a(+) plasmid, which has been used as an antigen in ELISA and WB for human toxoplasmosis diagnosis [47]. In the context of animal welfare, Cakir-Koc and colleagues used complete and incomplete Freund’s adjuvant, which causes severe local inflammation and other physiological changes that may harm the hens’ well-being and antibody production, despite its good adjuvant effect [61]. For this reason, we used chitosan as an adjuvant, a natural polymer derived from chitin, which can induce intrinsic immune recognition via PRRs by signaling the activation of adaptive immunity, with fewer adverse effects reported in animals compared to other adjuvants, and found good immunogenicity [62,63,64].

The IgY advantages and relatively easy production in hens further facilitate large-scale antibody production, a crucial consideration for developing diagnostic kits or passive immunization therapies [36,65,66]. On the other hand, these antibodies may be used to purify or capture SAG1 into diagnostic assays, such as ELISA, WB, and FC, to detect specific antibodies; with the advantage that IgY exhibits low background interference due to its inability to bind to mammalian Fc receptors, augmenting accuracy [67]. On the other hand, purified IgY recognized both recombinant and endogenous SAG1, in both denatured and native forms, and on tachyzoites, which highlights its potential for directly detecting the parasite in different experiments, such as in cell culture or even as a valuable diagnostic test for acute infection by detection of the parasite in blood.

IgY, or a part of this molecule, could also serve as a therapeutic agent, such as the chicken single-chain fragment variable (IgY-scFv), which functions as a functional fragment in biomedical applications [68]. The passive immunization approach, especially for vulnerable populations such as pregnant women, could lower the risk of acquired toxoplasmosis infection, particularly if IgY is administered orally, as previously reported to be effective prophylaxis against other infections acquired by this route [36,69]. Since IgY does not activate the complement system, it may reduce inflammatory responses associated with conventional antibody therapies and simultaneously neutralize the parasite [32,70]. This feature, along with its ability to specifically bind SAG1, suggests that IgY could be further explored as part of a novel passive immunization strategy against *T. gondii* [31,36,65,66]. Nevertheless, some characteristics must be considered before using IgY in mammalian applications. Its short half-life and lack of Fc receptor interactions limit persistence and effector functions, which is disadvantageous in chronic or immune-mediated therapies but can be beneficial in acute or localized treatments by reducing systemic effects [71]. Similarly, the absence of ADCC and complement activation hampers efficacy in cancer or viral clearance, but it also avoids inflammation-driven damage in other diseases. For our purposes, SAG1 was selected as an antigen, since its surface expression on tachyzoites allows direct neutralization by IgY, which does not require Fc-mediated activity [28].

There are studies suggesting that basal human antibodies against IgY may reduce its efficacy and that human immunoglobulins contain a significant fraction of IgG anti-chicken IgY, especially in individuals whose diet includes poultry as a crucial component [72,73]. Although extremely high doses of IgY may induce some adverse effects to consider (e.g., systemic immune response against it and hypersensitivity-like reactions, resembling serum sickness, in pigs [74,75,76,77]), there is insufficient data about IgY adverse outcomes in humans, as with any other biotherapeutic or pharmacological treatment at low or moderate doses.

The capacity of IgY to neutralize *T. gondii* and prevent infection in vivo clearly represents an area for further research. The characterization of anti-SAG1 IgY antibodies in this study presents opportunities for further investigation into their therapeutic applications in other parasitic diseases, thereby providing a broader scope for their use beyond toxoplasmosis.

## 5. Conclusions

In the present study, the coding region of the SAG1 protein from *T. gondii* was cloned and heterologously expressed. The rSAG1 protein was used to immunize hens, resulting in the production of IgY antibodies that reacted with high specificity to recombinant and native SAG1. Furthermore, future assays should be performed to support the potential of IgY as a valuable tool for further development in diagnostic platforms and exploratory therapeutic research targeting *T. gondii*. The less harmful and cost-effective production of IgY positions it as a useful alternative to traditional mammalian antibodies.

## Figures and Tables

**Figure 1 antibodies-14-00081-f001:**
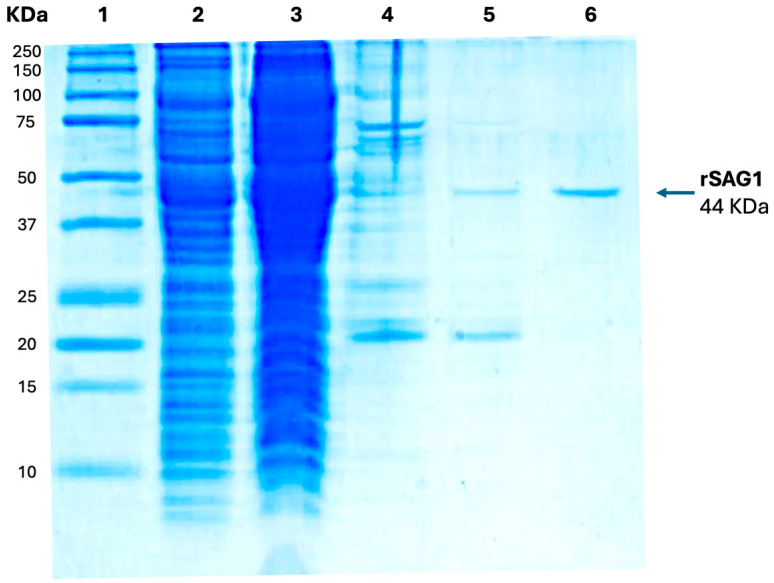
SDS–PAGE analysis of recombinant SAG1 (rSAG1) expression using the pET32a(+) vector in *E. coli*. fractions obtained during rSAG1 purification. Lane 1: Kaleidoscope Bio-Rad Marker; lane 2: *E. coli* crude extract; lane 3: flow-through protein fraction; lane 4: proteins obtained from the column washed with 25 mM imidazole; lane 5: proteins obtained from the column washed with 50 mM imidazole; lane 6: proteins obtained from the column washed with 75 mM imidazole.

**Figure 2 antibodies-14-00081-f002:**
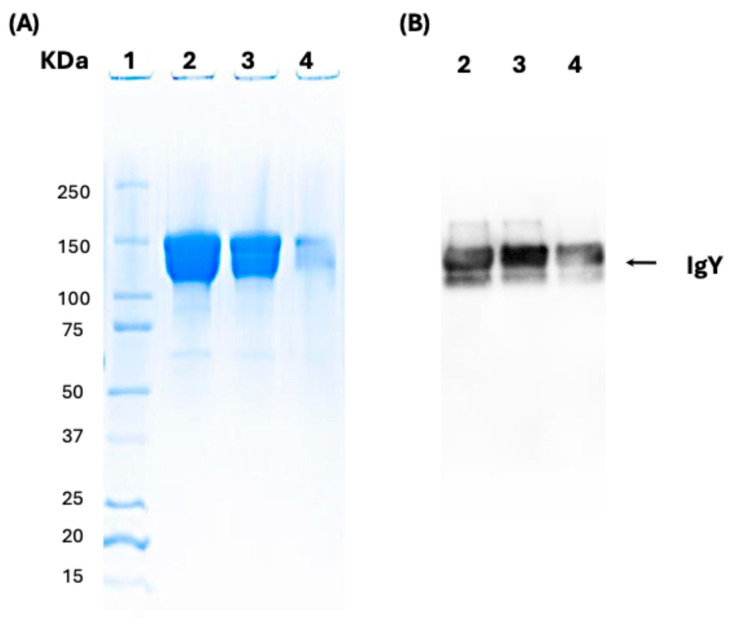
Characterization of IgY anti-rSAG1 by SDS-PAGE and Western Blot (WB). (**A**) Purification of IgY anti-SAG1 as detected in SDS-PAGE (**A**) and WB (**B**). A significant protein band corresponding to IgY with a molecular weight of approximately 150 kDa is observed. Lanes: 1. kaleidoscope Bio-Rad marker; 2. IgY 50 μg; 3. IgY 25 μg; 4. IgY 5 μg. (**B**) IgY immunodetection using rabbit antibody against IgY coupled to HRP. Lanes: 2. IgY 50 μg; 3. IgY 25 μg; 4. IgY 5 μg.

**Figure 3 antibodies-14-00081-f003:**
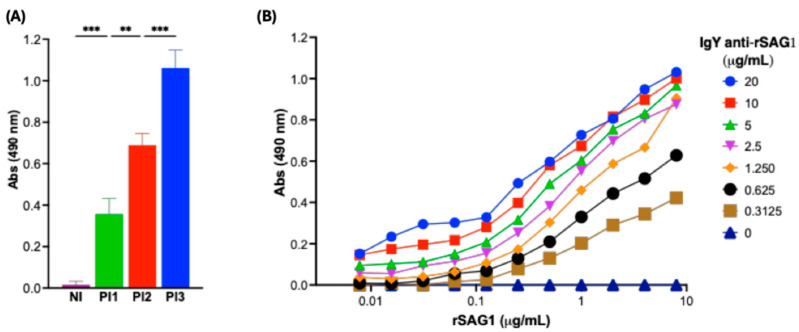
Specific detection of rSAG1 by IgY anti-SAG1 by indirect ELISA. Purified IgY anti-SAG1 antibodies were tested against rSAG1 at 8, 4, 2, 1, 0.5, 0.25, 0.125, 0.0625, 0.0313, 0.0156, and 0.0078 μg/mL. (**A**) The results are expressed as means ± standard deviation of triplicate assays. Asterisks indicate a significant difference among groups. Nl: Non-Immunized; Pl1: Post First Immunization 1; Pl2: Post Second Immunization 2; Pl3: Post Third Immunization 3. (** *p* < 0.01, and *** *p* < 0.001). The statistical analysis was performed using a one-way ANOVA with Tukey’s post hoc using GraphPad Prism version 10.3.1 for Mac, GraphPad Software, Boston, MA, USA, www.graphpad.com. (**B**) Purified IgY anti-SAG1 specifically recognizes rSAG1 in an antigen- and IgY concentration-dependent manner.

**Figure 4 antibodies-14-00081-f004:**
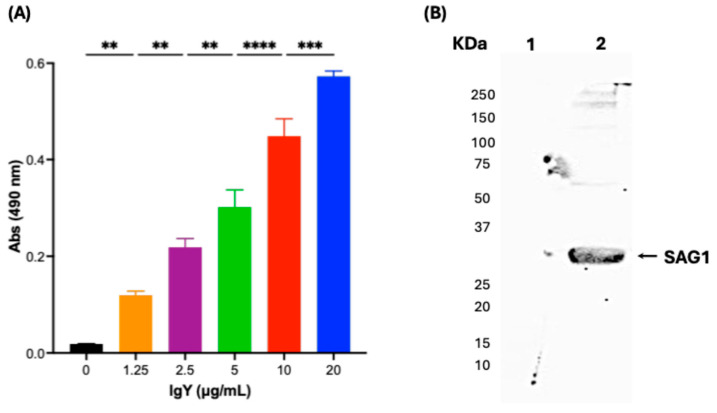
Specific detection of native and denatured endogenous SAG1 in *T. gondii* RH tachyzoite crude extract (TgCE) by IgY anti-SAG1. (**A**) Purified IgY anti-SAG1 specifically recognizes native endogenous SAG1 in a crude antigen- and IgY concentration-dependent manner. The TgCE was used at 2 μg/mL. The results are expressed as mean ± standard deviation of triplicate assays (** *p* < 0.01, *** *p* < 0.001, and **** *p* < 0.0001). The statistical analysis was performed using a one-way ANOVA with Tukey’s post hoc using GraphPad Prism version 10.3.1 for Mac, GraphPad Software, Boston, MA, USA, www.graphpad.com. (**B**) IgY produced against rSAG1 detects denatured endogenous SAG1 in TgCE. Line 1: Non-related protein crude extract from *E. coli* (5 μg), and Line 2: TgCE (5 μg) revealed with IgY anti-rSAG1.

**Figure 5 antibodies-14-00081-f005:**
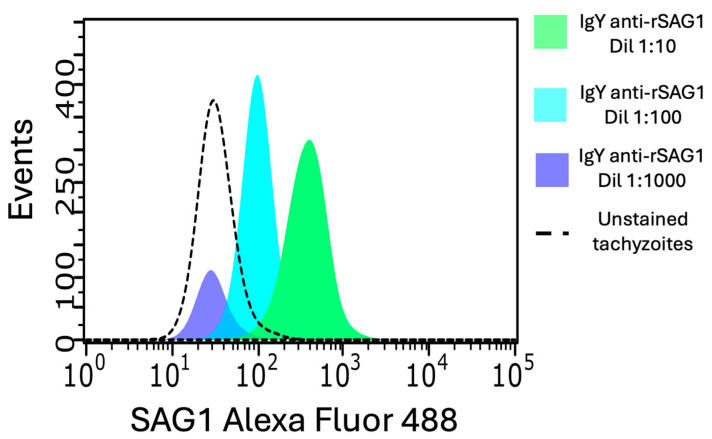
Specific detection of SAG1 in RH *T. gondii* tachyzoites. In-house fluorescently labeled IgY anti-SAG1, diluted at 1:10, 1:100, and 1:1000, was used to stain *T. gondii* RH tachyzoites and analyzed by flow cytometry.

## Data Availability

The original contributions presented in this study are included in the article. Further inquiries can be directed to the corresponding author.

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
