# Peer review of "Purification and Characterization of Immunoglobulin Y (IgY) Targeting Surface Antigen 1 (SAG1) of Toxoplasma gondii"

_2073-4468, 2025, doi:10.3390/antib14040081_

Round 1
Reviewer 1 Report
Comments and Suggestions for Authors
Purification and Characterization of Immunoglobulin Y (IgY) Targeting Surface Antigen 1 (SAG1) of Toxoplasma gondii
The methodology of the work is clearly described and involves cloning, expression, purification, immunization, IgY extraction and evaluation of specificity. Multiple validation methods (ELISA, Western blot, flow cytometry) were employed to show IgY recognizes both recombinant and native protein.
- What was the yield of IgY per egg.
- The conclusion that IgY could be useful for “diagnosis and treatment” seems like an over generalization. No diagnostic sensitivity/specificity data, no therapeutic efficacy models, and no in vivo results are shown.
3. Potential drawbacks of IgY include shorter half-life in mammals, limited effector functions, variability in egg yield, cross-reactivity risks, how does author work around these issues needs to be discussed in the discussion.
Author Response
- What was the yield of IgY per egg.
Thank you for asking. We included these results in section 3.2:
The average concentration of IgY per egg before and after the first immunization was 5.57 mg/mL. A noticeable increase was seen after the second and third immunizations, reaching an average concentration of 10.15 mg/mL. This gradual rise is consistent with the expected patterns of IgY production in laying hens, reported elsewhere.
- The conclusion that IgY could be useful for “diagnosis and treatment” seems like an over generalization. No diagnostic sensitivity/specificity data, no therapeutic efficacy models, and no in vivo results are shown.
We appreciate the reviewer’s observation.
The current study was designed as a first characterization of IgY anti-SAG1, and we agree that further experiments are necessary to substantiate its diagnostic and therapeutic potential. We are already in the process of obtaining approval from the Ethics and Animal Research Committees to evaluate SAG1-specific IgY using human sera and mouse models. These studies will allow us to determine diagnostic sensitivity and specificity, as well as to explore therapeutic applications in relevant in vivo settings. Thus, while the present manuscript focuses on the initial characterization of the antibody, the perspectives for its validation in diagnostics and therapy are part of our planned future work.
To better reflect the scope and limitations of our work, we revised our conclusion as follows:
“In the present study, the coding region of the SAG1 protein from T. gondii was cloned and heterologously expressed. The rSAG1 protein was used to immunize hens, resulting in the production of IgY antibodies that reacted with high specificity to recombinant and native SAG1. Furthermore, these findings support the potential of IgY as a valuable tool for further development in diagnostic platforms and exploratory therapeutic research targeting T. gondii. The less harmful and cost-effective production of IgY positions it as a useful alternative to traditional mammalian antibodies”.
We hope that this revised conclusion more accurately captures the contributions of our study, while also highlighting the need for future work to evaluate diagnostic performance and therapeutic efficacy.
- Potential drawbacks of IgY include shorter half-life in mammals, limited effector functions, variability in egg yield, cross-reactivity risks, how does author work around these issues needs to be discussed in the discussion.
We have added a discussion about these drawbacks in 4. Discussion section:
“Nevertheless, some characteristics must be considered before using IgY in mammalian applications. Its short half-life and lack of Fc receptor interactions limit persistence and effector functions, which is disadvantageous in chronic or immune-mediated therapies but can be beneficial in acute or localized treatments by reducing systemic effects [71]. Similarly, the absence of ADCC and complement activation hampers efficacy in cancer or viral clearance, but it also avoids inflammation-driven damage in other diseases. For our purposes, SAG1 was selected as an antigen, since its surface expression on tachyzoites allows direct neutralization by IgY, which does not require Fc-mediated activity [28]”.
Reviewer 2 Report
Comments and Suggestions for Authors Several studies have demonstrated that SAG1 protein from T. gondii is one candidate antigen for the diagnosis and vaccine development for toxoplasmosis. This manuscript addresses the production of the SAG1, which was cloned and heterologously expressed in E. coli. The rSAG1 protein was used to immunize hens to obtain IgY antibodies with high specificity to SAG1. The assays were performed with the recombinant protein and with parasites using flow cytometry analysis. The results assessed by flow cytometry on fixed tachyzoites demonstrated that Alexa Fluor 488 anti-SAG1 IgY diluted 1:100 and 1:10 can detect SAG1. However, the assays were performed with the parasite directly, and the antibody concentration needs to be high. What would the results be like for assays with T. gondii in cultured cell monolayers and paraffin-embedded mouse tissues? The potential of a diagnostic and therapeutic tool for toxoplasmosis needs to be evaluated by some experiment in this manuscript. Another option is to use sera to distinguish T. gondii-infected from non-infected individuals. Only the assays present in the current version of the manuscript do not support the conclusions presented about the potential of the antibody obtained. Minor issues 1. "IgY, or a part of this molecule, could also serve as a therapeutic agent or contribute to the development of one." The authors need to be clearer with the development of one. 2. Figure 3. Nl: Non-Immunized; Pl1: Post Immunization 1; Pl2: Post Immunization 2; Pl3: Post Immunization 3. It is not clear what each of the conditions is. It would be related to the protocol: "Immunization consisted of three injections into the breast muscle, each containing 150 μg of the rSAG1 antigen in 1% chitosan as an adjuvant, in a final volume of 0.2 mL. The second dose was administered two weeks after the first, followed by a third dose one week later." 3. Figure 4 does not have statistical analysis, while the caption for Figure 3 details the statistical analysis, the authors could include the software in which it was performed. 4. Figure 4. "Non-related protein crude extract from E. coli (5 ug)." It is necessary to make clear how this extract was obtained.Author Response
- Several studies have demonstrated that SAG1 protein from gondiiis one candidate antigen for the diagnosis and vaccine development for toxoplasmosis. This manuscript addresses the production of the SAG1, which was cloned and heterologously expressed in E. coli. The rSAG1 protein was used to immunize hens to obtain IgY antibodies with high specificity to SAG1. The assays were performed with the recombinant protein and with parasites using flow cytometry analysis. The results assessed by flow cytometry on fixed tachyzoites demonstrated that Alexa Fluor 488 anti-SAG1 IgY diluted 1:100 and 1:10 can detect SAG1. However, the assays were performed with the parasite directly, and the antibody concentration needs to be high. What would the results be like for assays with T. gondii in cultured cell monolayers and paraffin-embedded mouse tissues? The potential of a diagnostic and therapeutic tool for toxoplasmosis needs to be evaluated by some experiment in this manuscript. Another option is to use sera to distinguish T. gondii-infected from non-infected individuals. Only the assays present in the current version of the manuscript do not support the conclusions presented about the potential of the antibody obtained.
R1.
We thank the reviewer for this observation. We acknowledge that proposing IgY as useful for “diagnosis and treatment” may be premature in the absence of data on diagnostic sensitivity and specificity, therapeutic efficacy, or in vivo validation. The present work was conceived as a first step, focusing on the molecular cloning of SAG1, the production of recombinant antigen, and the generation of specific IgY antibodies. We agree that further experiments are required to establish their true diagnostic and therapeutic value. Thus, this manuscript is an initial characterization, with subsequent validation forming part of ongoing and future work.
To better reflect the scope and limitations of our work, we revised our conclusion as follows:
“In the present study, the coding region of the SAG1 protein from T. gondii was cloned and heterologously expressed. The rSAG1 protein was used to immunize hens, resulting in the production of IgY antibodies that reacted with high specificity to recombinant and native SAG1. Furthermore, these findings support the potential of IgY as a valuable tool for further development in diagnostic platforms and exploratory therapeutic research targeting T. gondii. The less harmful and cost-effective production of IgY positions it as a useful alternative to traditional mammalian antibodies”.
Minor issues 1. "IgY, or a part of this molecule, could also serve as a therapeutic agent or contribute to the development of one." The authors need to be clearer with the development of one. We have modified this idea as follows: IgY, or a part of this molecule, could also serve as a therapeutic agent, such as the chicken single-chain fragment variable (IgY-scFv), which functions as a functional fragment in biomedical applications [68]. 2. Figure 3. Nl: Non-Immunized; Pl1: Post Immunization 1; Pl2: Post Immunization 2; Pl3: Post Immunization 3. It is not clear what each of the conditions is. It would be related to the protocol: "Immunization consisted of three injections into the breast muscle, each containing 150 μg of the rSAG1 antigen in 1% chitosan as an adjuvant, in a final volume of 0.2 mL. The second dose was administered two weeks after the first, followed by a third dose one week later." We have modified according to the reviewer's suggestions. 3. Figure 4 does not have statistical analysis, while the caption for Figure 3 details the statistical analysis, the authors could include the software in which it was performed. We have added the reviewer's suggestions. 4. Figure 4. "Non-related protein crude extract from E. coli (5 ug)." It is necessary to make clear how this extract was obtained. We have modified section 2.3, now titled as: Recombinant SAG1 (rSAG1) and E. coli crude extract
Reviewer 3 Report
Comments and Suggestions for Authors
Most things are fine. It should be useful by measuring and addressing the value of Kd.
Author Response
1. Most things are fine. It should be useful by measuring and addressing the value of Kd.
R1.
Based on your suggestion, we calculated Kd according to Beatty et al in section 2.6 and section 3.3:
“Furthermore, the dissociation constant (Kd) was calculated following the method described by Beatty et al., yielding a value of 2.22 × 10⁻⁹ M, which indicates a high-affinity interaction between IgY anti-SAG1 and SAG1”.
Round 2
Reviewer 2 Report
Comments and Suggestions for Authors
The authors made the suggestions, but did not add results to indicate diagnostic and therapeutic value. In this sense, they changed the statements on the topic, but the conclusion sentence needs to be corrected.
"Furthermore, these findings support the potential of IgY as a valuable tool for further development in diagnostic platforms and exploratory therapeutic research targeting T. gondii" change to "Furthermore, future assays should be performed to support the potential..."
In addition, in "Figure 5. Specific detection of SAG1 in RH T. gondii tachyzoites." the MFI values should be described in the manuscript text to support the differences between the concentrations shown in the Figure.
Author Response
We thank the reviewer for their valuable comments and suggestions.
In response:
-
We have revised the conclusion sentence as recommended.
-
Regarding Figure 5, we have now included the MFI values in the manuscript text to support the differences between the concentrations shown in the figure. These values are described in the Results section to provide a clearer interpretation of the data.
We appreciate the reviewer’s insights, which have helped us improve the clarity and scientific rigor of our manuscript.
Reviewer 3 Report
Comments and Suggestions for Authors
It seems to be sufficient to be published.
Author Response
We thank the reviewer for their valuable comments and suggestions.